# Effectiveness of cash-plus programmes on early childhood outcomes compared to cash transfers alone: A systematic review and meta-analysis in low- and middle-income countries

Madison T. Little[1,2]*, Keetie Roelen[3], Brittany C. L. Lange[1,4], Janina I. Steinert[1,5], Alexa R. Yakubovich[1,6], Lucie Cluver[1,7], David K. Humphreys[1,2]

1 Department of Social Policy & Intervention, University of Oxford, Oxford, United Kingdom, 2 Green Templeton College, University of Oxford, Oxford, United Kingdom, 3 Centre for Social Protection, Institute for Development Studies, Brighton, United Kingdom, 4 Child Health & Development Institute of Connecticut, Farmington, Connecticut, United States of America, 5 School of Governance, Technical University of Munich, Munich, Germany, 6 MAP Centre for Urban Solutions, Unity Health Toronto & University of Toronto, Toronto, Canada, 7 Department of Psychiatry & Mental Health, University of Cape Town, Cape Town, South Africa

* madison.little@spi.ox.ac.uk

## Abstract

### Background

To strengthen the impact of cash transfers, these interventions have begun to be packaged as cash-plus programmes, combining cash with additional transfers, interventions, or services. The intervention's complementary ("plus") components aim to improve cash transfer effectiveness by targeting mediating outcomes or the availability of supplies or services. This study examined whether cash-plus interventions for infants and children <5 are more effective than cash alone in improving health and well-being.

### Methods and findings

Forty-two databases, donor agencies, grey literature sources, and trial registries were systematically searched, yielding 5,097 unique articles (as of 06 April 2021). Randomised and quasi-experimental studies were eligible for inclusion if the intervention package aimed to improve outcomes for children <5 in low- and middle-income countries (LMICs) and combined a cash transfer with an intervention targeted to Sustainable Development Goal (SDG) 2 (No Hunger), SDG3 (Good Health and Well-being), SDG4 (Education), or SDG16 (Violence Prevention), had at least one group receiving cash-only, examined outcomes related to child-focused SDGs, and was published in English. Risk of bias was appraised using Cochrane Risk of Bias and ROBINS-I Tools. Random effects meta-analyses were conducted for a cash-plus intervention category when there were at least 3 trials with the same outcome. The review was preregistered with PROSPERO (CRD42018108017). Seventeen studies were included in the review and 11 meta-analysed. Most interventions operated

**Data Availability Statement:** All relevant data are within the manuscript and its Supporting Information files.

**Funding:** The authors received no specific funding for this work.

**Competing interests:** The authors have declared that no competing interests exist.

**Abbreviations:** BCC, behaviour change communication; BFP, Bolsa Família Program; CCT, conditional cash transfer; cRCT, cluster-randomised controlled trial; FHP, Family Health Program; LMICs, low- and middle-income countries; SDGs, Sustainable Development Goals; UCT, unconditional cash transfer; WASH, water, sanitation, and hygiene.

during the first 1,000 days of the child's life and were conducted in communities facing high rates of poverty and often, food insecurity. Evidence was found for 10 LMICs, where most researchers used randomised, longitudinal study designs ($n = 14$). Five intervention categories were identified, combining cash with nutrition behaviour change communication (BCC, $n = 7$), food transfers ($n = 3$), primary healthcare ($n = 2$), psychosocial stimulation ($n = 7$), and child protection ($n = 4$) interventions. Comparing cash-plus to cash alone, meta-analysis results suggest Cash + Food Transfers are more effective in improving height-for-age (d = 0.08 SD (0.03, 0.14), $p = 0.02$) with significantly reduced odds of stunting (OR = 0.82 (0.74, 0.92), $p = 0.01$), but had no added impact in improving weight-for-height (d = −0.13 (−0.42, 0.16), $p = 0.24$) or weight-for-age z-scores (d = −0.06 (−0.28, 0.15), $p = 0.43$). There was no added impact above cash alone from Cash + Nutrition BCC on anthropometrics; Cash + Psychosocial Stimulation on cognitive development; or Cash + Child Protection on parental use of violent discipline or exclusive positive parenting. Narrative synthesis evidence suggests that compared to cash alone, Cash + Primary Healthcare may have greater impacts in reducing mortality and Cash + Food Transfers in preventing acute malnutrition in crisis contexts. The main limitations of this review are the few numbers of studies that compared cash-plus interventions against cash alone and the potentially high heterogeneity between study findings.

## Conclusions

In this study, we observed that few cash-plus combinations were more effective than cash transfers alone. Cash combined with food transfers and primary healthcare show the greatest signs of added effectiveness. More research is needed on when and how cash-plus combinations are more effective than cash alone, and work in this field must ensure that these interventions improve outcomes among the most vulnerable children.

### Author summary

#### Why was this study done?

- Cash transfers (providing individuals or families direct cash payments) are an easy-to-implement intervention that has widespread impacts, but evidence suggests that these programmes do not universally improve child health and well-being.

- Cash-plus programmes (combining cash transfers with complementary interventions) have been proposed as a solution to maximise the effectiveness of cash transfers to improve the lives of children.

- Our study aimed to assess whether cash-plus programmes are more effective than cash alone in improving child health and wellbeing.

## What did the researchers do and find?

- We conducted a systematic review of 42 information sources and databases and found 17 studies that met the review criteria, of which 11 were meta-analysed.

- We identified 5 cash-plus programme categorisations: Cash + Nutrition Behaviour Change Communication, Cash + Food Transfers, Cash + Primary Healthcare, Cash + Psychosocial Stimulation, and Cash + Child Protection.

- Meta-analysis results suggest that only Cash + Food Transfers has added impact above cash alone, having significantly reduced odds of children experiencing stunted growth (OR = 0.82 (0.74, 0.92)).

- Narrative synthesis results suggest that Cash + Food Transfers in crisis contexts and Cash + Primary Healthcare may also have greater benefit than cash alone.

## What do these findings mean?

- There are few studies to date that evaluate the effectiveness of cash-plus programmes against cash alone, which leaves significant evidence gaps in our understanding of these interventions.

- Our findings suggest that not all cash-plus programme combinations are more effective than cash transfers alone but that combining cash with food transfers or primary health-care may have added impact in improving child health and well-being.

- There was significant variation in impacts across studies and because of the limited number of studies identified for analysis, more research is needed in identifying effective plus-components and effective models of how these cash-plus programmes are designed and implemented.

## Introduction

Compared to adults, children are disproportionately affected by poverty and its consequences, with nearly one-fifth of all children living in extreme poverty [1]. Social protection interventions, including cash transfers (direct monetary provision), aim to mitigate the risk and effects of poverty and social exclusion [2]. Cash transfers often do not have restrictions on how the cash is spent but may or may not have conditional requirements to receive the transfer (e.g., regular child growth monitoring). A well-established evidence base indicates that cash transfers improve many aspects of children's lives, including increased food security and improved school attendance [3]. Nevertheless, impacts have been more mixed and less overwhelming in more challenging areas of child development, including nutrition [4] and health [5].

In efforts to strengthen the impact of these interventions, cash transfers have begun to be packaged as cash-plus programmes, combining cash transfers with other interventions or services (e.g., behaviour change communication (BCC), psychosocial support, or cross-sectoral linkages) [6]. Cash-plus programmes can be classified as multisectoral interventions that seek to improve development outcomes rather than a time-restricted grant to exit poverty [7].

These interventions are argued to be more effective than cash alone because the "plus" component specifically targets factors that are necessary for cash transfers to have impact but that cash alone does not change (e.g., improved mediating outcomes and/or availability of supplies or services). Multisectoral interventions such as cash-plus programmes are conceptually at the forefront of efforts to improve outcomes for infants and young children. Notably, the Nurturing Care Framework for Early Childhood Development proposes an integrated response for young children across health, nutrition, child protection, social protection, and education sectors [8].

Intervening in early childhood is vital for moral, economic, and social reasons. Ensuring that children can reach their full development potential means that they have better health and higher economic earnings as adults, and, in turn, reduce intergenerational transmission of poverty [9]. The multisectoral nature of cash-plus programmes speaks to multiple Sustainable Development Goals (SDGs). Cash transfers are central to poverty alleviation (SDG 1) and in achieving national social protection floors (SDG Target 1.3). The plus-components aim to promote child development through other mechanisms than poverty alleviation, such as through direct nutrition support (SDG 2), health services (SDG 3), early child education (SDG 4), or child protection (SDG 16). The 2 elements of cash-plus programming link across multiple SDGs, which is supported by frameworks that emphasise that achieving each SDG is interdependent with the achievement of other SDGs [10]. This interdependence suggests that multisectoral interventions can have a greater impact than vertical, siloed interventions alone.

Cash-plus interventions hold much promise in accelerating achievement of multiple SDG targets for children. However, to the best of our knowledge to date, there has been no synthesis to evaluate if these multisectoral interventions are more effective than cash transfers alone.

This review sought to answer the question: Are cash-plus interventions for infants and children under the age of 5 more effective than cash transfers alone in improving child health and well-being outcomes across the SDGs?

It is important to emphasise that this review is not comprehensive of all cash-plus programmes for young children. Rather, this review assesses the effectiveness of cash-plus interventions compared to cash alone as opposed to a pure (no-intervention) control. It aims to provide evidence on whether and when combining cash with plus-components can have an accelerating impact in achieving child-focused SDGs.

## Methodology

This paper presents a systematic review and meta-analysis for cash-plus interventions compared to cash transfers alone, which are targeted to families with infants and children in low- and middle-income countries (LMICs). This study is reported as per the Preferred Reporting Items for Systematic Reviews and Meta-Analyses (PRISMA) guidelines (S1 PRISMA Checklist) [11]. The protocol was preregistered with PROSPERO (CRD42018108017).

### Information sources and search

For this review, we searched 11 electronic databases, 27 grey literature sources, and 4 trial registries. Four journals were hand-searched, and 10 experts in the field of development economics and cash-plus interventions were contacted to identify unpublished literature. The information sources and sample search strategy are provided in S1 Text. All searches were completed through 06 April 2021. The search strategy was informed by literature on the review topic and contains categories for the regions and individual countries, the population (infants and children), and terms for social protection interventions, which were replicated and modified from Owusu-Addo and colleagues [12].

## Eligibility criteria

**Population/Participants.** The study had to evaluate an intervention package implemented in one or more LMICs, as defined by the World Bank. The intervention package had to aim to improve outcomes (below) among infants and young children aged 0 to 59 months and be implemented during this age period.

**Intervention.** The intervention package had to contain at least 2 components. The first component requirement was a cash transfer intervention (SDG 1.3) that met 4 requirements [12], specifically that the programme (1) provide financial assistance at the individual or household level; (2) be noncontributory (i.e., individuals have not paid into the system) and in the form of a nonrepayable, unrestricted grant (i.e., no requirement for how cash was used); (3) aim to reduce the impacts of, or vulnerability to, poverty (monetary or multidimensional); and (4) be disbursed in consistent, predictable intervals.

The second component requirement was having at least 1 "plus" intervention targeting SDG 2 (No Hunger), SDG 3 (Good Health and Well-being), SDG 4 (Education), or SDG 16 (Violence Prevention). Specific plus-components include nutrition support programmes (i.e., food transfers and BCC) to reduce malnutrition (SDG 2.1 to 2.2), interventions to control communicable diseases and reduce infant/child mortality (SDG 3.2 to 3.3), coverage with health insurance (SDG 3.8), early childhood development, care, and preprimary education (SDG 4.2), or violence prevention and parenting interventions (SDG 16.2).

Preliminary searches found that water, sanitation, and hygiene (WASH) interventions were integrated as part of these plus-components (rather than as stand-alone plus-interventions), thus were not assessed as its own plus-component. No restriction was placed on behavioural conditions for receiving cash (i.e., both unconditional and conditional cash transfers were included). Studies comparing the effects of unconditional cash transfers (UCTs) versus conditional cash transfers (CCTs) were excluded because condition monitoring was not conceptualised as a plus component.

**Comparison.** Studies had to include at least (1) one group receiving the cash-plus intervention and (2) one group receiving cash-only. Because this review aimed to evaluate the added benefit of the plus-component to cash transfers, studies that evaluated no treatment versus cash-plus only (without having a separate cash-only group) were excluded.

**Outcomes.** Cash transfers alone have had limited impact on distal outcomes for child well-being (e.g., on nutritional status and health [4,5]), despite strong evidence that cash transfers can improve more proximal outcomes, such as improving access to food [3,13,14]. Thus, cash-plus programmes were developed in hopes of achieving improvements in these distal, "third-order" outcomes. This review concentrates on these third-order outcomes that contribute to SDG indicators related to children under 5. These include measures of poverty (including multidimensional poverty), malnutrition (including stunting, wasting, underweight, and obesity), morbidity or mortality (neonatal, infant, and children under 5) including from unsafe water or lack of sanitation/hygiene or infectious disease, psychosocial and cognitive development, and violence against children. Within the included studies that examine these outcomes of interest, we also comment on the impact of proximal outcomes and their contribution in the causal pathway.

**Time.** Studies conducted between 2000 and 2021 were included. The start year (2000) reflects the year that the Millennium Development Goals were adopted, which began a global roll out of evidence-based interventions, including cash transfers. Additionally, pairing cash transfers to another intervention is new conceptual thinking that is not likely to have been investigated prior to the start date.

**Study design.**   Included studies had to have used an experimental or quasi-experimental design. Studies that did not measure the outcome at both baseline and post-intervention were excluded.

**Other exclusion criteria.**   Studies not published in English, interventions related to pregnancy/childbirth, interventions on farming/agricultural productivity practices, or microfinance/savings interventions.

## Study selection and data extraction

Studies were imported into Rayyan and deduplicated prior to screening [15]. Studies were double screened blind to reduce potential bias. Initial agreement between raters was >96%, and reviewers agreed on the final included studies. When multiple publications were available for the same evaluation (i.e., baseline, midline, and endline reports), the endline report is cited, and when endline findings are published in both report and journal article form, the journal article was cited. A standardised data extraction form was used, with categories for population characteristics and context, intervention design and components, outcomes and effects, pathways, and equity evidence [16–19]. Two authors extracted meta-analysis data independently to minimise errors.

## Risk of bias

Assessing for risk of bias at the study level, randomised studies were evaluated using the Cochrane Risk of Bias Tool and nonrandomised studies using the ROBINS-I tool [20,21]. All studies were double coded, and disagreements discussed. Piloting of the risk-of-bias assessment on 6 studies had 100% agreement between raters and >95% for the full set of studies. Visualisations were created using *robvis* software [22]. The quality assessment was used to comment on the strength and limitations of the evidence base and the confidence in recommendations from the synthesis [23]. Due to a limited number of retrieved studies across a range of cash-plus combinations and outcomes, no meaningful analysis could be conducted to assess for publication bias.

## Data synthesis

Studies were meta-analysed when there were at least 3 trials for an outcome with similar follow-up times for the same cash-plus intervention; this included outcomes in Cash + Nutrition BCC, Cash + Food Transfers, Cash + Psychosocial Stimulation, and Cash + Child Protection. In instances when a study used a pure (no-intervention) control with 2 treatment arms (cash-only and cash-plus), results were reanalysed using the cash-only arm as the control. The analysis corrected for clustering to address possible unit of analysis error by adjusting the sample size as necessary [24]. Specifically, in cases when studies allocated treatment at the cluster level but presented standard errors that were not adjusted for this clustering, the effective sample size was used in analysis. This value was calculated by dividing the original sample size by the design effect, which is computed using the average cluster size and intraclass correlation [25].

Standardised mean differences (d) for continuous outcomes (i.e., cognitive development and anthropometric z-scores) and log odds ratios for binary outcomes (i.e., anthropometric outcome, violence against children, and positive parenting) were calculated before running the random-effects meta-analyses in R using the Knapp and Hartung adjustment, which accounts for few data points [26–28]. In addition to anthropometric z-score values, odds ratios were calculated from the proportion of children experiencing stunting, wasting, and underweight status. Cognitive development effect sizes were calculated from cognition subscales (Bayley Scale of Infant Development) or aggregate cognitive scale score (McCarthy Scales of Children's

Abilities–General Cognitive Index); general measures of child development (Ages and Stages Questionnaire) were synthesised narratively. Parenting practices were assessed in meta-analyses using the prevalence of harsh discipline and exclusive positive parenting. Results from meta-analyses are displayed using forest plots and heterogeneity results using funnel plots (S1 Fig). Quantitative measures of consistency/heterogeneity are reported ($I^2$ and $\tau^2$). Confidence intervals for $I^2$ are also provided for context on the uncertainty of the value [29]. However, there were too few studies to explore heterogeneity further.

Study effects were synthesised narratively when there were too few studies to meta-analyse. When studies were narratively synthesised, effect sizes were transformed to cash-only versus cash-plus comparisons. Intervention design was classified as 1 of 4 models set forth in a previous review of Cash + Parenting Programmes [30]. Quantitative equity effects in impact were noted, defined as either subgroup analyses or interaction effects [31].

## Results

### Study selection

From 5,097 unique articles identified in the search, 80 full-text articles were reviewed for inclusion. Sixty-three were excluded (see Fig 1 and S2 Text); the majority were excluded either because the study did not have a cash-only group or the study did not meet the criteria for a cash-plus programme. Eleven protocols were identified that could meet the criteria for inclusion in an update to the review; study teams were contacted, who all confirmed no intervention impact data were yet available. Seventeen studies were included in the review, of which 11 were meta-analysed (Fig 1 and S2 Text). The studies included in this review represent 11 unique cash-plus programmes.

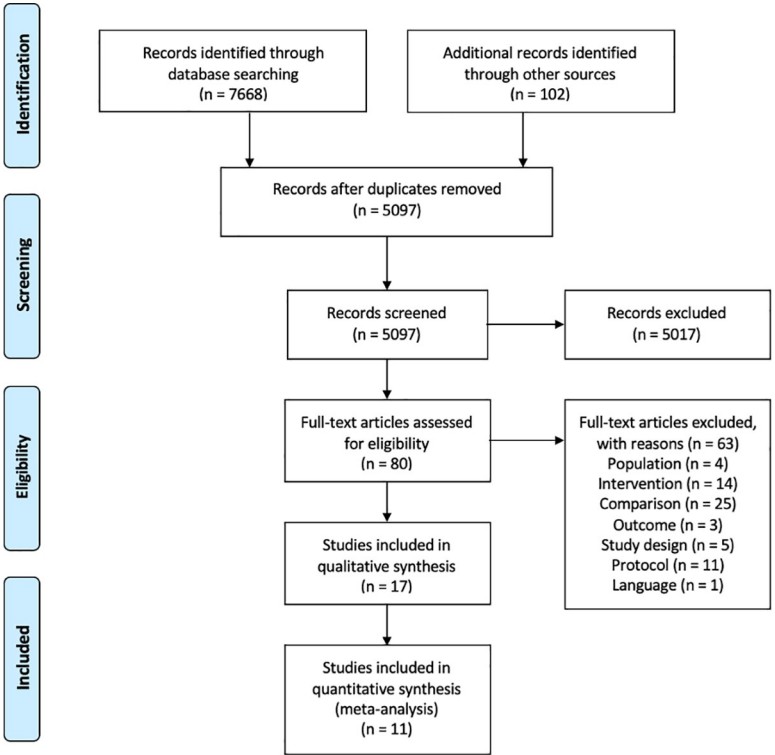

**Fig 1. PRISMA flow chart.**

## Study characteristics

The majority of studies focus on the first 1,000 days of a child's life and cite the critical importance of intervening during this period to maximise development potential. The studies included in this review cover countries in sub-Saharan Africa (7), Latin America (6), and South and Southeast Asia (4). CCTs (6 studies), UCTs (7 studies), and public works programmes (4 studies) were studied; all CCTs were implemented in Latin America, where that type of transfer is prominent. In addition to the cash transfer, 5 categories of "plus" components were identified: nutrition BCC ($n = 7$), food transfers ($n = 3$), primary healthcare ($n = 2$), psychosocial stimulation ($n = 7$), and child protection interventions ($n = 4$).

There are 2 dominant theories about how these cash-plus programmes work: (1) The package works by improving mediating outcomes on the pathway to impact (e.g., maternal knowledge of proper nutrition) while also addressing the structural deprivations impacting health and development potential [32]; and (2) Supply-side and demand-side interventions must occur in tandem to meet population needs (e.g., providing health services when health checks are a cash condition) [33,34].

Of the 9 studies that reported cash values, standardised transfer amounts (converted from local currency to US dollars at the time of intervention) ranged from approximately $8/month to $75/month and were often distributed monthly. The plus-components were delivered either monthly or weekly. In providing cash value justifications, studies noted either matching values from other cash transfer initiatives in the country or setting the value based on the country's poverty level and cost of necessary household expenses. When calculating the latter value, interventions covered up to 20% of household expenses [35].

Three out of the 4 intervention designs described by Arriagada and colleagues [30] were identified in the included studies. Six programmes (8 studies) utilised an integrated design, specifically that the plus-component was nested within, and operated by, the cash transfer programme. A further 4 programmes (7 studies) had a convergence design, whereby the 2 components were implemented separately but there was explicit coordination between implementing partners. This contrasts to one programme (2 studies) that used an alignment design, in which there was separate implementation of the 2 components and no coordination between implementers (regardless of whether the same population was reached). No studies were identified that utilised a piggybacking design, which would have the cash component delivered within an already-existing plus intervention.

Only one programme [36–38] was designed for active father involvement throughout the intervention period. Three other programmes [32,39–42] had a portion of the plus component dedicated to father education, of which one programme [40,41] also invited mothers-in-law to participate in hopes of creating a supportive household environment. Further, only 4 programmes exclusively used a home visit model [36–38,43–45].

All studies were done as cluster-randomised controlled trials (cRCTs) except for one randomised at the individual level [44] and 3 using a quasi-experimental design [33,34,46]. Studies were published from 2013 to 2021 and had follow-up times ranging from 4 months to 4 years post-intervention. Details of individual interventions are shown in Table 1 below, and further information is available in S1 Table. Table 2 provides an overview map of the review findings based on the intervention category and outcome.

## Risk of bias of individual studies

Risk of bias was low for most studies, emphasising a high-quality evidence base and strong methodological rigour. One study was rated as having some concerns in risk of bias because of deviations from the planned intervention, specifically in implementation challenges that led to

**Table 1. Intervention components and outcomes.**

| Study | CT and Intervention Design | Nutrition BCC | Food Transfer | Primary Healthcare | Psychosocial Stimulation | Child Protection |
|---|---|---|---|---|---|---|
| **Langendorf 2014** | *Unconditional CT (Integrated)* | | Anthropometrics, mortality | | | |
| **Field 2021** | *Unconditional CT (Convergence)* | Anthropometrics, child illness | | | | |
| **Khan 2019** | *Unconditional CT (Convergence)* | Anthropometrics | Anthropometrics | | | |
| **Guyatt 2018** | *Unconditional CT (Integrated)* | Anthropometrics | | | | |
| **Ahmed 2019** | *Unconditional CT (Integrated)* | Anthropometrics, child illness | Anthropometrics, child illness | | | |
| **Ahmed 2020** | *Unconditional CT (Integrated)* | Poverty | | | | |
| **UNICEF 2020** | *Cash-for-Work (Integrated)* | Anthropometrics | | | | |
| **Premand 2020** | *Unconditional CT (Integrated)* | Anthropometrics | | | Child development | Violence against children; positive parenting; child illness |
| **Barnhart 2020** | *Cash-for-Work (Convergence)* | | | | Child development | Violence against children; positive parenting |
| **Betancourt 2020** | *Cash-for-Work (Convergence)* | | | | | Violence against children; positive parenting; child illness |
| **Jensen 2021** | *Cash-for-Work (Convergence)* | | | | Child development | Violence against children |
| **Attanasio 2014** | *Conditional CT (Integrated)* | | | | Child development, anthropometrics | |
| **Andrew 2018** | *Conditional CT (Integrated)* | | | | Child development | |
| **Fernald 2016** | *Conditional CT (Convergence)* | | | | Child development | |
| **Kagawa 2017** | *Conditional CT (Convergence)* | | | | Child development | |
| **Guanais 2013** | *Conditional CT (Alignment)* | | | Post-Neonatal Infant Mortality | | |
| **da Silva 2019** | *Conditional CT (Alignment)* | | | Child Mortality | | |

BCC, behaviour change communication; CT, cash transfer.

difficulties in individuals accessing cash [44]. All randomised trials followed children longitudinally throughout the trial period. The 3 quasi-experimental studies were rated as having moderate risk of bias because although the studies account for confounding, there is still greater risk of bias than if the studies could have used a randomised design. Similarly, these 3 studies relied on repeat cross-sectional data, which introduce some bias in potentially uneven exposure to the intervention by assessing within-population change rather than within-person change. No study was rated as having high risk of bias. Individual study assessments are available in S3 Text.

## Synthesis of results

**Cash + Nutrition Behaviour Change Communication (BCC).** Seven studies were identified that combined cash transfers with nutrition BCC [32,40,41,44–47]. Most BCC

**Table 2. Overview map of findings from meta-analyses and synthesis.**

| Intervention | Outcome | Cash vs Cash-Plus |
|---|---|---|
| *Cash + Nutrition BCC* | | |
| | Stunting | **No difference (6)** |
| | Wasting | **No difference (6)** |
| | Underweight | **No difference (6)** |
| | Fever | Mixed findings (2) |
| | Diarrhoea | No difference (2) |
| | Cough/Cold | Mixed findings (2) |
| | General Child Illness | Cash-Plus more effective (1) |
| | Poverty | Cash-Plus more effective (1) |
| *Cash + Food Transfer* | | |
| | Wasting (short-term crisis) | **Cash-Plus more effective (1)** |
| | Stunting (long-term impact) | **Cash-Plus more effective (2)** |
| | Wasting (long-term impact) | **No difference (2)** |
| | Underweight (long-term impact) | **No difference (2)** |
| | Child Illness | No difference (1) |
| | Mortality | Cash-Plus more effective (1) |
| *Cash + Primary Healthcare* | | |
| | Mortality | Cash-Plus more effective (2) |
| *Cash + Psychosocial Stimulation* | | |
| | Cognitive Development | **No difference (3)** |
| | Overall Child Development | Cash-Plus more effective (1) |
| *Cash + Child Protection* | | |
| | Use of Harsh Discipline | **No difference (3)** |
| | Exclusive Positive Parenting | **No difference (3)** |
| | Child Illness | Mixed findings (3) |

Meta-analysis findings are bolded.

BCC, behaviour change communication.

components covered UNICEF's Essential Family Practices on maternal and child nutrition, including exclusive breastfeeding (age <6 months) and complementary feeding (age >6 months), health and hygiene practices, and use of health services preventively and promptly when a child becomes ill [47]. Three anthropometric indicators were assessed (height-for-age, weight-for-height, and weight-for-age) as well as the odds of children experiencing stunting, wasting, or being underweight, respectively, which is defined as having z-scores $\leq-2$ standard deviations of the WHO Child Growth Standards Median [48].

Meta-analysis findings (Fig 2) suggest that cash-plus programmes are not more effective than cash transfers alone in reducing odds of children experiencing stunting (OR = 0.95 (95% CI 0.83, 1.09), $p$ = 0.40; $I^2$ = 26% (95% CI 0, 87)), wasting (OR = 0.99 (0.93, 1.05), $p$ = 0.64; $I^2$ = 0% (0, 14)), or underweight status (OR = 1.01 (0.84, 1.20), $p$ = 0.93; $I^2$ = 17% (0, 98)). Additional meta-analysis findings (Fig 3) also suggest no added impact on z-score measures of anthropometrics for stunting (d = 0.03 (−0.04, 0.09), $p$ = 0.36, $I^2$ = 24% (0, 88)), wasting (d = −0.03 (−0.12, 0.07), $p$ = 0.51, $I^2$ = 47% (0, 93)), or being underweight (d = −0.03, (−0.11, 0.05), $p$ = 0.45, $I^2$ = 31% (0, 92)). Sensitivity analyses were also conducted by removing the quasi-experimental study [46] from the meta-analyses. No sensitivity test showed a significant z-score change for height-for-age (d = 0.04 (−0.03, 0.11), $p$ = 0.18), weight-for-height (d = −0.03 (−0.15, 0.09), $p$ = 0.57), or weight-for-age (d = −0.02 (−0.13, 0.08), $p$ = 0.61).

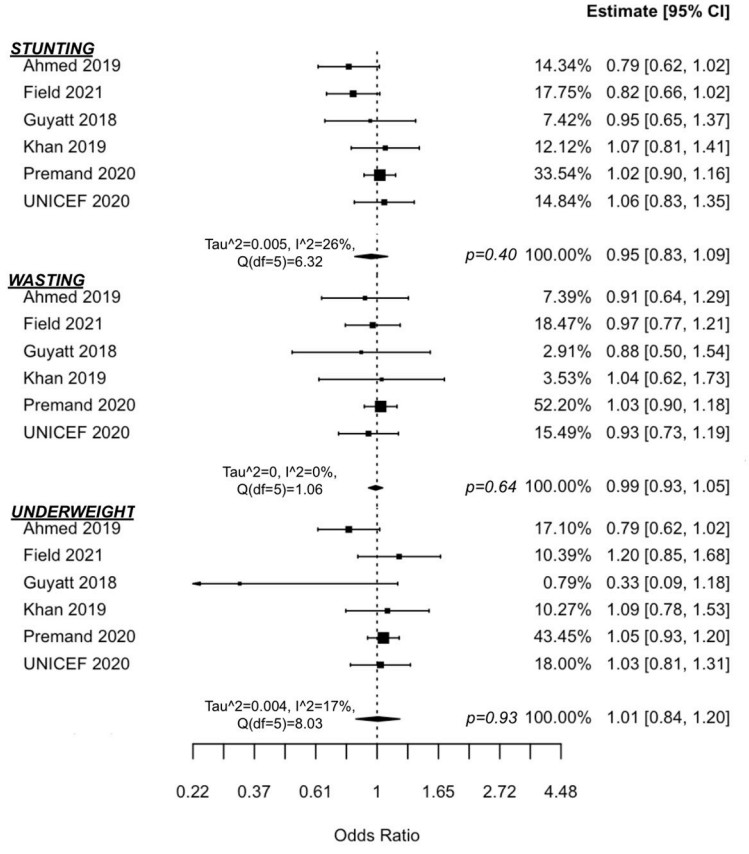

**Fig 2. Forest plot of Cash + Nutrition BCC vs. cash alone on anthropometric odds ratios.**

One study in Bangladesh found no gendered differences in anthropometric impacts [40], while another in Niger found that impact on wasting was driven by improvements in boys [47]. Compared to cash alone, the intervention effects of the cash-plus intervention in Kenya for wasting/underweight reduction were greater in smaller household sizes and greater for wealthier households in reducing stunting [44]. Another study in Myanmar found the cash-plus intervention to be significant in decreasing moderate stunting compared to cash alone, with no impact on severe stunting [32].

There is also evidence that Cash + Nutrition BCC may reduce poverty. At the end of the intervention in Bangladesh [41], researchers found that cash-plus had greater reduced poverty head count (20% lower) and depth (6% lower) and severity (3% lower) of poverty compared to cash alone. While these differences were no longer statistically significant 4 years post-programme, there is other evidence of sustained reductions in poverty. First, while many families moved out of poverty during the programme, approximately 40% in both cash-only and cash-plus groups fell back into poverty by the 4-year follow-up. However, families in the cash-plus arm experienced greater movement out of poverty during the intervention period than those receiving cash alone; thus at 4 years post-evaluation, there was a 10-percentage point difference between cash-plus and cash-only groups in the proportion of families that had exited poverty and stayed nonpoor. The authors theorise that this was contributable to the evidence that the plus-component further enhanced women's social capital and agency. Second, 4 years after the intervention ended, there was an added 16% reduction in chronic monetary poverty from cash-plus compared to cash alone, as measured by the Calvo-Dercon Poverty Score [49].

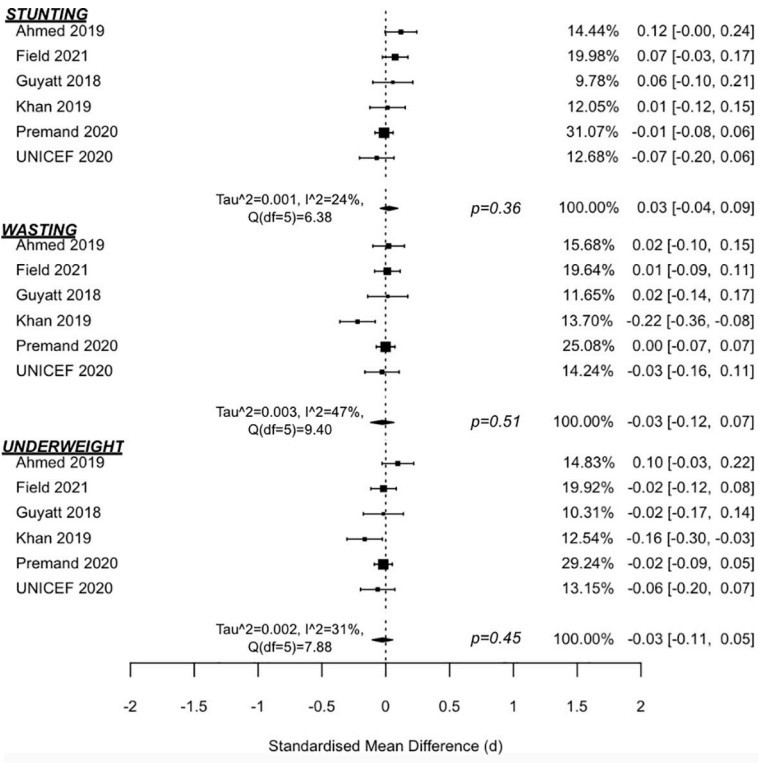

**Fig 3. Forest plot of Cash + Nutrition BCC vs. cash alone on anthropometric z-scores.**

Multiple indicators along the causal pathway between cash and child anthropometrics showed improvement of cash-plus over cash alone. The most notable is improvements in WASH practices (5/5), including improved defecation (2/2), handwashing (4/5), and treating water (1/1) [32,40,44,45,47]. One study also found significant reductions in iron deficiency anaemia (1/1) [45].

However, several other indicators demonstrated no added benefit of cash-plus. This includes in increasing birth registration (0/2) [46,47], complete immunisation (0/2) [44,47], or in reducing vitamin A deficiency (0/2) [45,47] or diarrhoea (0/2) [32,40].

Markers of food security were mixed, specifically for increased food consumption (2/2) [32,40], improved dietary diversity scores or minimally acceptable diet (3/5) [32,40,44,46,47], or increasing number of feedings (0/1) [40]. In contrast, one additional study found that cash-only had better outcomes than the cash-plus group on dietary diversity, iron food consumption, minimum meal frequency, and minimum acceptable diet [45]. There were also mixed findings on increasing ever or early breastfeeding (2/4) [32,44,46,47], increasing women's decision-making agency (1/2) [32,41], increasing medical care seeking for a sick child (1/2) [32,47] or reducing fever and cough/cold (1/2) or general child illness (1/1) [32,40,47].

**Cash + food transfer.** Three studies aimed to reduce or prevent undernutrition [40,45,50], all by supplementing cash with additional food transfers. Studies were conducted in impoverished regions with high rates of food insecurity and undernutrition, where approximately half of all study children had stunted growth [40,45,50].

One study was implemented during an acute crisis in the hunger gap in Niger when rates of undernutrition, diarrhoeal diseases, and malaria increase [50]. The trial compared cash alone to cash-plus 1 of 3 different food transfer formulations. The 3 trial arms were meta-analysed to

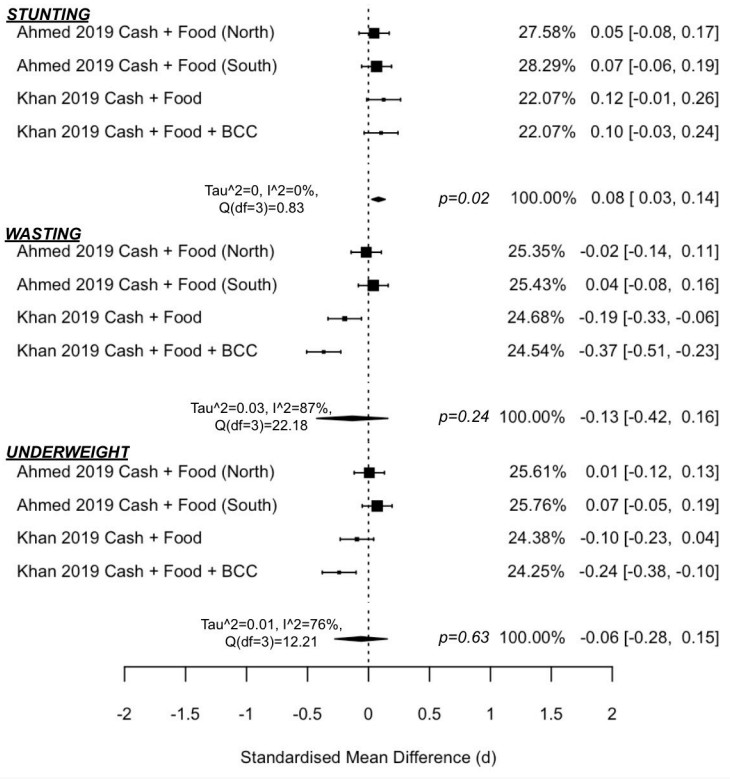

**Fig 4. Forest plot of Cash + Food Transfers vs. cash alone on anthropometric z-scores.**

calculate an overall impact of the 3 cash-plus arms in preventing global acute malnutrition. Compared to cash alone, children in households receiving Cash + Food had significantly reduced odds of experiencing acute malnutrition (OR = 0.41 (0.27, 0.63), $p$ = 0.012). The cash-plus groups had up to 8 times lower mortality incidence rates than cash alone. When children died, the leading suspected cause was malaria (76%) and gastroenteritis (14%).

Meta-analysis results of studies in long-term development contexts (Figs 4 and 5) suggest that cash-plus is more effective over cash alone in increasing height-for-age z-score (d = 0.08 (0.03, 0.14), $p$ = 0.02, $I^2$ = 0% (0, 74)), which translated to significantly reduced odds of children experiencing stunting (OR = 0.82 (0.74, 0.92), $p$ = 0.01, $I^2$ = 0% (0, 70)). There was no added impact in improving weight-for-height z-score (d = −0.13 (−0.42, 0.16), $p$ = 0.24, $I^2$ = 87% (57, 99)) or weight-for-age z-score (d = −0.06 (−0.28, 0.15), $p$ = 0.43, $I^2$ = 76% (24, 98)). Similarly, there was no added impact in reducing odds of children experiencing wasting (OR = 0.89 (0.70, 1.14), $p$ = 0.24, $I^2$ = 0% (0, 86)) or underweight status (OR = 0.93, (0.80, 1.09), $p$ = 0.26, $I^2$ = 0% (0, 84)), respectively.

The analysis included 2 studies, one in Pakistan with 2 separate trial arms (both receiving Cash + Food, but one also receiving BCC) [45] and one in Bangladesh, which had 2 separate trials (one each in the north-western and southern regions of the country) [40]. To note, this latter study set the cash-plus value to equal the cash-only value (i.e., the cash-plus group received half the cash value of the cash-only transfer) as opposed to providing the same cash value to both groups, but also supplementing with food transfers. There was no difference between cash-only and Cash + Food in reducing fever, cough/cold, or diarrhoea in either the North or South trial in Bangladesh [40].

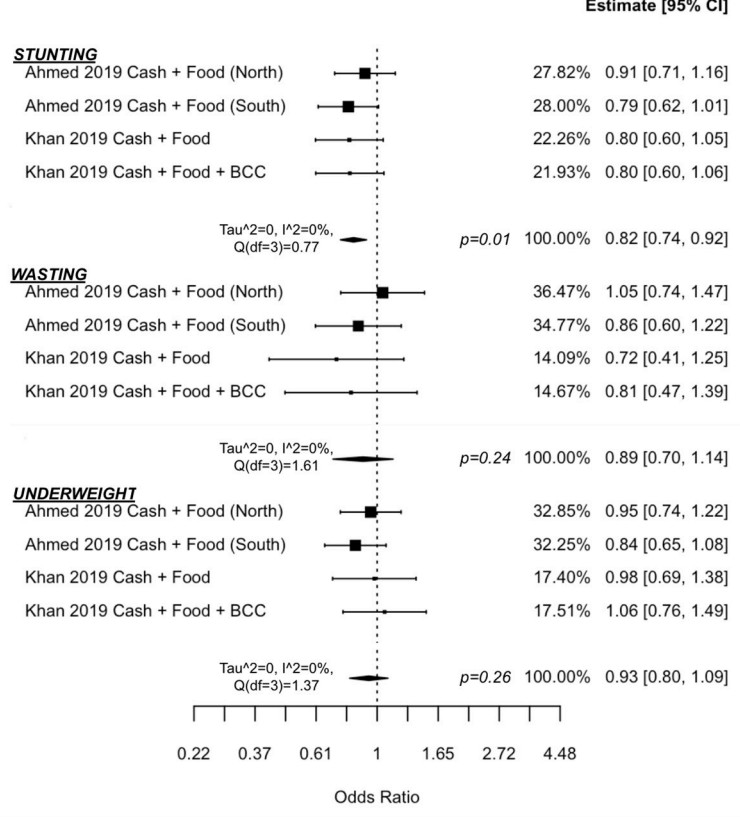

**Fig 5. Forest plot of Cash + Food Transfers vs. cash alone on anthropometric odds ratios.**

**Cash + primary healthcare.** Only 2 studies were identified that evaluated provision of cash and primary healthcare services (e.g., growth monitoring, preventive health services, or medical treatment for illness). Both were quasi-experimental studies that examined Brazil's independent scale-ups of the CCT, Bolsa Família Program (BFP), and primary healthcare, Family Health Program (FHP), which provided care from community health workers to doctors. Although the programmes were each scaled up separately (an alignment design), the findings from both studies demonstrate the interdependence of the 2 programmes in achieving success in reducing post-neonatal infant [33] and child [34] mortality rates (both measured as deaths per 1,000 live births). Both study equations showed that BFP and FHP each had statistically significant, independent effects in reducing mortality rates. Guanais [33] also included a statistically significant interaction term for the 2 programmes, indicating that there is also an interdependence of the 2 programmes in further reducing infant mortality. For example, in a municipality with coverage rates for BFP and FHP at 25% and 0%, respectively, the predicted mortality rate was 5.24 deaths/1,000 live births as opposed to coverage of 60% and 100% having a rate of 1.38 death/1,000 live births [33]. Combining cash and primary healthcare had the greatest impact in the highest poverty regions, and this is especially important given the high inequality rate in Brazil [33]. The models indicate that the impact of FHP is higher at higher coverage levels of BFP, thus highlighting the need to scale up the demand-side and supply-side interventions in concert [33].

**Cash + psychosocial stimulation.** Seven studies, representing 4 unique programmes, evaluate the impacts of cash-plus programming on measures of child development. One

programme used group-based social and BCC methods of which child psychosocial development is one of many topics covered [47], one used group-based parenting support [39,42], and 2 programmes exclusively used a home visit model [36,38,43,51].

Studies employed a variety of child development measures, of which 3 included specific subscales or scale measures of cognitive development. After standardising the measures, meta-analysis findings (Fig 6) suggest that these Cash + Psychosocial Stimulation programmes may not be more effective than cash transfers alone in promoting overall cognitive development (d = 0.16 (−0.25, 0.57), $p$ = 0.24, $I^2$ = 85% (47, 100)), although there is substantial heterogeneity among the studies.

Although a pilot study was underpowered to detect improvements in child development [36], the main trial in Rwanda found significant improvements in overall child development from cash-plus over cash alone (d = 0.21 SD (0.09, 0.33)) [38]. The trial in Niger also found added improvements in socioemotional development (using the strength-and-difficulties questionnaire) for children receiving cash-plus over cash alone (d = 0.149 SD (0.03, 0.27)) [47]. One study also tested the addition of micronutrient supplementation, but found no added benefit for cognitive development or child growth [43].

Even for interventions that are found to be effective, however, caution must be raised for the longer-term effect of the intervention package. While one cash-plus programme showed positive effects above cash alone in the short-term evaluation (d = 0.26 SD improvement in cognitive scores in Colombia) [43], the effects dissipated within 2 years after the intervention [51]. This likewise applies to anticipated mediators of child development. Three out of the 4 programmes found significant improvements across the variety of play materials and variety and frequency of play activities [36,37,43,47,52]. However, in the same 2-year post-intervention follow-up study in Colombia, the intervention impacts on these mediators were no longer statistically significant [51].

Comparing different socioeconomic subgroups in cash-plus arms to cash alone, these intervention packages varied in effects. While there was evidence of higher cognitive impact for Mexican indigenous communities in the stratified sample and among children with lowest scores at baseline, the intervention effects also favoured wealthier households [39]. Two studies in Colombia and Mexico found the interventions favoured mothers with greater education [39,51].

Only one study in our review was identified that evaluated 2 different design models against a cash-only control [39], which found that only Mexican children in the convergence intervention design had improved cognitive development and not in the alignment design. Secondary analysis was also done to assess if impacts differed when women became pregnant [42]; the

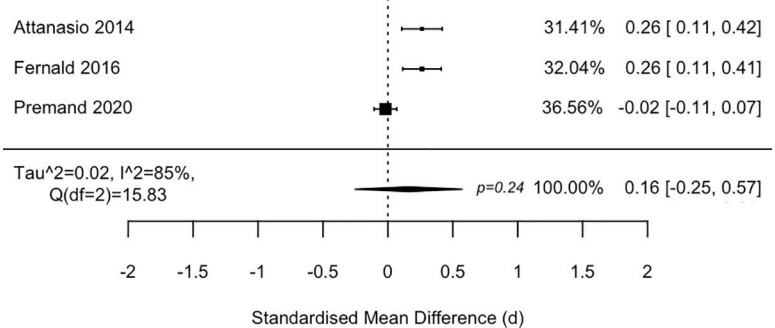

**Fig 6. Forest plot of Cash + Psychosocial Stimulation vs. cash alone on cognitive development.**

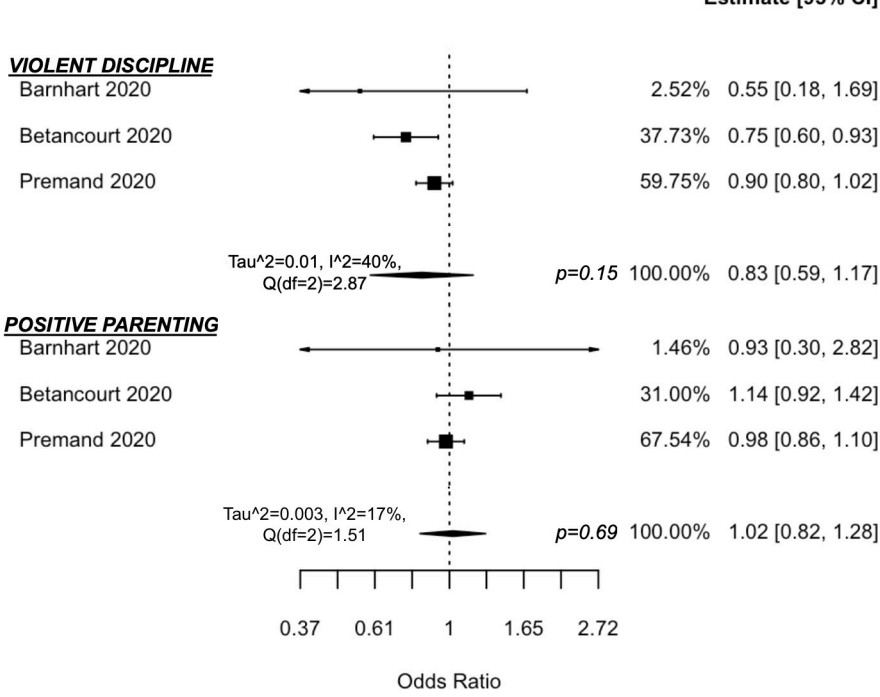

**Fig 7. Forest plot of Cash + Child Protection vs. cash alone on parenting practices.**

study found that only mothers in the convergence intervention design who became pregnant after adolescence had improved child development.

**Cash + child protection.**   Four studies of 2 unique programmes [36–38,47] were designed using a comprehensive early child development framework, with child protection as one of the main pillars. Both programmes used home visits, but Premand and Barry [47] also had small-group and village assembly components in Niger. A pilot evaluation [36] notes that the larger effectiveness trial [37,38] also included "booster sessions" in efforts to prolong and sustain intervention effects in Rwanda. Studies were done in high poverty and food insecure contexts, and one study found very high levels (almost 50% prevalence) of violent discipline to children and one-third of mothers had experienced violence in the last 3 months [37]. Violent discipline was measured by the prevalence of any harsh discipline from parents to child (e.g. forbidding, shaking, yelling, spanking with/without an object, berating, slapping, and hitting hands with/without an object) and exclusive positive parenting measured by the absence of these discipline methods and, instead, disciplining through explaining or redirecting the child [47].

Although both main programme trials found reductions in violent discipline, meta-analysis findings suggest that Cash + Child Protection is not more effective than cash alone in reducing parental violent discipline of children (OR = 0.83 (0.59, 1.17), $p$ = 0.15, $I^2$ = 40% (0, 99)) or increasing exclusive use of positive parenting (OR = 1.02 (0.82, 1.28), $p$ = 0.69, $I^2$ = 17% (0, 96)) (Fig 7). One study found that the decline in parental violent discipline was driven by reductions in subscales on the parental use of shaking, spanking, berating, slapping, and hand-hitting, but there was no impact on the use of yelling, forbidding, or hitting their child with objects [47].

The programme in Rwanda found that in addition to reductions in violent discipline of children, there was added benefit of cash-plus above cash alone in increasing father

engagement in childcare, improving shared decision-making, and reducing maternal depression and female violence victimisation, but no added benefit in reducing male violence perpetration [36–38]. The intervention effects on reduced violent discipline were maintained 1 year post-intervention (incidence rate ratio (IRR) = 0.74 (0.66, 0.84), $p < 0.001$) [38]. There was no difference in impact for boys and girls.

These comprehensive early child development programmes also evaluated the impact on childhood illness. Despite an almost 2 times higher odds of accessing clean water in the cash-plus group compared to cash-only group at endline, Betancourt and colleagues [37] found no additional impact from the cash-plus intervention on childhood illness, including diarrhoea, fever, and cough, though this may be due to the change from rainy (baseline) to dry (endline) seasons in Rwanda; the pilot study had only found reductions in cough prevalence compared to cash alone [36]. Premand and Barry [47], however, found a 19% reduction in the cash-plus group compared to cash alone in the child being sick in the past month in Niger.

## Discussion

### Summary of the evidence

This systematic review and meta-analysis investigated whether cash-plus programmes were more effective than cash alone in accelerating outcome improvements in young children. In addition to the evidence synthesis, this review contributes to the growing literature of situating cash transfers as the "control" or standard of care. This is particularly relevant to studies assessing the cost-effectiveness of new interventions and their value compared to if a low-cost cash transfer programme were implemented as opposed to nothing [53,54].

The review found cash-plus programmes with 5 types of plus-components that are focused on young children. These cash-plus programmes linked cash transfers for poverty alleviation (SDG 1) with SDG 2 (Nutrition BCC, Food Transfers), SDG 3 (Primary Healthcare), SDG4 (Psychosocial Stimulation), and SDG 16 (Child Protection). These intervention packages also touch all 5 areas of the multisectoral Nurturing Care Framework for Early Child Development, namely by integrating cash (Security and safety) with efforts to drive (1) Adequate nutrition; (2) Good health; (3) Opportunities for early learning; and (4) Responsive caregiving [8]. Given the large focus in global health on psychosocial stimulation to improve child development and BCC to address undernutrition, it is unsurprising that these 2 interventions represented the majority studies. Nonetheless, each of the intervention categories was limited in the total number of studies identified.

Meta-analysis results concluded that Cash + Food Transfers may be more effective than cash alone in reducing stunting. However, meta-analysis findings suggest no added impact above cash alone in Cash + Nutrition BCC for improving anthropometrics, Cash + Psychosocial Stimulation for improving cognitive development, or Cash + Child Protection for reducing violent discipline or increasing exclusive positive parenting. However, there was potentially substantial heterogeneity across the meta-analyses. The narrative synthesis found preliminary evidence that Cash + Primary Healthcare may have greater impacts than cash alone in reducing mortality, Cash + Food Transfers may have greater impacts than cash alone in reducing acute malnutrition and mortality in crisis contexts, Cash + Nutrition BCC may have greater impacts than cash alone in reducing poverty, and Cash + Child Protection trials suggest a trend towards greater impact than cash alone in reducing violent discipline.

Overall, the studies included in this review had low risk of bias, and while the studies were rigorously conducted, there are still a very limited number of studies examining cash-plus versus cash alone, and therefore, the ability to generalise is limited until more studies are published. This review found that cash-plus programmes are being employed throughout LMICs

and more than two-thirds of the included studies were published in the last 3 years. As this is an emerging area in social protection, there is hope that more research will become available in the coming years that evaluate the interventions against cash controls, some of which have been published as protocols (see S2 Text). This review is timely and especially relevant as new cash-plus programmes continue to be planned and scaled up [55].

A previous review of cash transfer programmes found that the evidence of cash impact decreases with each step further in the causal pathway between cash input and changes in child health and well-being [56]. To address the limitations in impact of cash transfers, cash-plus programmes were developed in hopes of reaching and improving these distal, "third-order" outcomes (child health and development), which our review investigated. Noting the limited evidence of impact on these outcomes, we also examined possible pathway contributions of cash-plus over cash alone within these studies. Assessment of causal pathways suggests that cash-plus likewise has little added benefit over cash alone, except for a few noteworthy differences such as in Cash + Nutrition BCC improving WASH practices and reducing poverty, Cash + Psychosocial Stimulation improving child stimulation practices and materials, and Cash + Child Protection reducing intimate partner violence. However, these results are in context of studies that evaluated the distal outcomes, thus there may be selection bias in these intermediary impact findings.

A recent meta-analysis of cash transfers (including cash transfers both with and without Nutrition BCC) found very small, but statistically significant improvements on stunting height-for-age z-scores (0.03+/−0.03 SD) compared to a no-intervention control, which accounts for a 2.1% reduction in stunting prevalence [4]. Among studies delivering Cash + Nutrition BCC, there was a statistically significant 3.1% reduction in stunting prevalence compared to a no-intervention control, but no impact on height-for-age z-score. Our review adds to the evidence base by specifically comparing Cash + Nutrition BCC to cash alone, finding that there was no statistically significant difference between the 2 interventions in reducing stunting; this may suggest that any improvements in stunting reduction are likely driven by the cash component rather than the added Nutrition BCC. In both our review and the review of cash transfers alone, there were very few studies that assessed child illness and more research is needed to assess the impact on this outcome and its contribution to the causal pathway between cash and child growth.

Lastly, there are limited and mixed impacts on vulnerable subgroups, having potential unintended consequences. Only about half of the included studies provided any equity evidence (defined as subgroup or interaction effects), of which only 3 studies conducted sex-disaggregated analyses; findings demonstrated either no difference in intervention effects between boys and girls (2) or the intervention effects favoured boys only (1). Few studies specifically involved fathers, which may place increased caregiving burden on mothers, such as collecting food/cash transfers, attending intervention sessions, or complying with conditions of cash transfers; this has also been identified as an issue in graduation programmes for early child development [57]. More work is needed to investigate and ensure that these interventions benefit the most vulnerable children, thereby narrowing the health equity gap (SDG 10).

## Limitations

The main limitation of this review is the few study numbers that were retrieved for each cash-plus programme. Findings from this review should be viewed as preliminary evidence and serve as a guide for future research, particularly given the potentially high heterogeneity in the meta-analyses. Ultimately, more research is needed before definitive conclusions can be made

on whether cash-plus is more effective than cash alone in improving outcomes for young children.

Because this is a review comparing cash-plus against cash alone, trials were not included that had only 2 arms (a cash-plus group and a no-intervention group), some of which are found in S2 Text. Thus, this systematic review retrieved a subset of all studies on cash-plus programmes, and notable programmes, such as Ghana's Cash + Health Insurance [58,59], were not included due to comparison criteria requirements. However, the goal of this review was to assess whether cash-plus programmes are more effective than cash alone rather than compared to no intervention. In addition to the cash-plus programmes included in this review, there are other combination interventions that may have an impact on children; examples include productivity and livelihood interventions that are targeted to households or adults, but have a benefit to improving child outcomes [60], or broader environmental health interventions such as water and sanitation management [61].

Some studies noted inherent difficulties in measurement (e.g., measuring change in standardised z-scores for anthropometrics) and that improvements in outcomes (e.g., stunting) may take longer than the intervention/study duration [44,50]. This review also comments on the impact of cash-plus on the causal pathways to the outcomes of interest. While the discussion of the intermediary outcomes provides context for the impact of cash-plus programmes compared to cash alone, there is a risk of selection bias in only discussing the findings from studies that also examined the distal outcomes and not including studies that only evaluated the impact of cash-plus on the intermediary outcomes.

Due to language abilities, this study only examined English studies; this may have also introduced selection bias through missing articles published through non-English outlets [62]. However, where known national programmes were operating or being planned, experts working on these programmes were contacted to identify potentially missed publications. Only one study was identified as being omitted by the search [63]. The cognitive impacts of this Cash + Psychosocial Stimulation intervention were published in English literature and included in this review [39]. However, the report also included nutritional impacts that were not reported in English.

### Recommendations for research

Our review identified a variety of study designs, which impact the conclusions that the individual studies can make on the effectiveness of a cash-plus programme. First, several studies used a cash control when large-scale cash transfer programmes were already operating in-country. Second, other studies used a no-intervention control and had cash-only and cash-plus arms; only a subset of these studies included additional analyses using the cash-only arm as the control to assess for the added impact of the plus-intervention above cash alone. Lastly, 2 studies used a natural experiment to assess the individual component impacts and their interaction. Although introducing more bias because of nonrandom intervention assignment, this design allowed for assessment of cash-plus at scale and its impact on population-level child outcomes.

An ideal study design would be a longitudinal, cRCT with 4 arms (control, cash-only, plus-only, and cash-plus) and which includes analyses using no-intervention as the control as well as each of the component parts. This would provide a complete picture of the impact of cash-plus programmes and the potential synergies of cash and plus components above and beyond either component alone. Only 3 programmes included in this review utilised a study design that would allow for most of these conclusions [40,41,47,50]. However, whether a no-intervention control is possible is impacted by whether a national cash transfer programme already exists and similarly, if no intervention would be unethical, such as in a crisis context.

There were high levels of heterogeneity in the meta-analyses and limited study numbers, thus more studies are needed before overall effectiveness can be determined. Promising areas of investigation include combining cash with food transfers, primary healthcare, and parenting interventions for child protection. Although Cash + Primary Healthcare was found to be effective in Brazil, there was a clear link between the 2 interventions (i.e., cash was conditioned on use of preventive health services). It is unclear if connecting access to health services and poverty alleviation interventions would be as effective if the transfer was unconditional. Other specific evidence gaps include no evaluations having been done in the Middle East and North Africa and East Asia and Pacific regions.

Two other essential areas for future investigation include (1) assessing for long-term effects of the intervention packages and (2) whether the effects are maintained when brought to scale. One study found some indicators of maintained poverty reductions 4 years post-programme [41], while another study noted that the lack of long-term impact on cognitive development was possibly due to implementation challenges at scale as opposed to a controlled and smaller efficacy trial [51].

Lastly, the heterogenous findings of studies included in this review emphasise the need to further investigate the role of implementation and context, cash-plus models, and selection of the plus-component in optimising the impact of cash-plus programmes. For example, only one programme was identified in the review that evaluated multiple cash-plus models, finding differential impacts based on design [39,42]. Only 3 programmes were identified that tested different plus-components in addressing the same child outcomes, each demonstrating that not every relevant plus-component will further improve outcomes for children [40,41,43,45]. Only one study was found to explicitly consider context (i.e., functioning or accessible food markets) in selecting the intervention package [40]. No included studies tested different intensities of the same plus-component. Optimising programme design, selection of the plus-component, and its implementation and intensity—in addition to the overall evidence of effectiveness (the scope of this review)—will be essential in improving outcomes for the most vulnerable children and maximising the cost-effectiveness of these programmes.

## Conclusions

Cash transfers alone may not achieve the outcome improvements in children that practitioners and policymakers aim to address, and, in these cases, cash-plus programming may be considered. While debates remain on the optimal package of interventions, this review provides preliminary evidence that the added plus-component in cash-plus may be more effective than cash alone when combining cash with food transfers to prevent acute malnutrition in crises and to reduce stunting in the long-term, primary healthcare to reduce mortality, nutrition BCC to reduce long-term poverty, and child protection interventions to reduce violent discipline. Findings also suggest that cash-plus is not more effective than cash alone when combined with nutrition BCC to improve child anthropometrics. Ultimately, more research is needed on how to optimise the promise of cash transfers and multisectoral intervention packages for young children to achieve the SDGs by 2030.

## Supporting information

**S1 Checklist. PRISMA Checklist.** PRISMA, Preferred Reporting Items for Systematic Reviews and Meta-Analyses.
(DOC)

**S1 Fig. Funnel plots.** Funnel plots provided for each meta-analysis.
(TIF)

**S1 Table. Study characteristics.** Information provided on country, participants, follow-up period, study design, cash amount, intervention intensity, intervention provider, and plus-intervention descriptions for each study.
(DOCX)

**S1 Text. Information sources and sample search strategy.** Sources of information searched and sample search strategy are provided.
(DOCX)

**S2 Text. Excluded studies at full text and list of included studies.** Table of excluded studies at full-text screening, with explanation, and a list of the studies included in the review.
(DOCX)

**S3 Text. Risk of bias assessments.** Individual study assessments for risk of bias using Cochrane Risk of Bias and ROBINS-I tools.
(DOCX)

**S4 Text. Study protocol.** Study protocol as published on PROSPERO.
(PDF)

## Acknowledgments

We would like to thank John Hoddinott and Elizabeth Maffioli as well as research teams from the World Bank (PI: Patrick Premand), Aga Khan University (PI: Sajid Soofi), Boston College (PI: Theresa Betancourt), Kimetrica Ltd (PI: Helen Guyatt), and UNICEF Kenya for generously providing us with supplemental data for the meta-analyses; the team at the Instituto Nacional de Salud Pública (PI: Lynnette Neufeld) for providing supplemental study information; and Mark Fransham for providing input on the meta-analyses. This work is part of Madison Little's doctoral research, and he is generously funded by the Economic & Social Research Council (UK), Green Templeton College (University of Oxford), and Clarendon Scholarship (University of Oxford). We would also like to thank the thousands of children and families that participated in the primary studies as we work to achieve a world in which all children survive and thrive.

## Author Contributions

**Conceptualization:** Madison T. Little, Keetie Roelen, Lucie Cluver, David K. Humphreys.

**Data curation:** Madison T. Little, Janina I. Steinert, Alexa R. Yakubovich.

**Formal analysis:** Madison T. Little, Brittany C. L. Lange.

**Investigation:** Madison T. Little.

**Methodology:** Madison T. Little, Keetie Roelen.

**Project administration:** Madison T. Little.

**Supervision:** Keetie Roelen, Lucie Cluver, David K. Humphreys.

**Validation:** Brittany C. L. Lange.

**Visualization:** Madison T. Little.

**Writing – original draft:** Madison T. Little.

**Writing – review & editing:** Madison T. Little, Keetie Roelen, Brittany C. L. Lange, Janina I. Steinert, Alexa R. Yakubovich, Lucie Cluver, David K. Humphreys.

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
