## [Editor Report · Decision Letter 0]

16 Feb 2021

Dear Dr Little, 

Thank you for submitting your manuscript entitled "Do cash-plus programmes improve early childhood outcomes more than cash transfers or child development interventions alone? A systematic review and meta-analysis in low- and middle-income countries." for consideration in PLOS Medicine’s Special Issue on Global Child Health.

Your manuscript has now been evaluated by the PLOS Medicine editorial staff as well as by the Special Issue Guest Editors and I am writing to let you know that we would like to send your submission out for external peer review.

Kind regards,

Caitlin Moyer, Ph.D.

Associate Editor

PLOS Medicine

---

## [Decision Letter · Decision Letter 1]

1 Apr 2021

Dear Dr. Little,

Thank you very much for submitting your manuscript "Do cash-plus programmes improve early childhood outcomes more than cash transfers or child development interventions alone? A systematic review and meta-analysis in low- and middle-income countries." (PMEDICINE-D-21-00432R1) for consideration in PLOS Medicine’s Special Issue on Global Child Health.

Your paper was evaluated by a senior editor and discussed among all the editors here. It was also discussed with the special issue Guest Editors, and sent to three independent reviewers, including a statistical reviewer. The reviews are appended at the bottom of this email and any accompanying reviewer attachments can be seen via the link below:

[LINK]

In light of these reviews, I am afraid that we will not be able to accept the manuscript for publication in the journal in its current form, but we would like to consider a revised version that addresses the reviewers' and editors' comments. Obviously we cannot make any decision about publication until we have seen the revised manuscript and your response, and we plan to seek re-review by one or more of the reviewers. 

We expect to receive your revised manuscript by . Please email us (plosmedicine@plos.org) if you have any questions or concerns.

We look forward to receiving your revised manuscript. 

Sincerely,

Caitlin Moyer, Ph.D.

Associate Editor 

PLOS Medicine

plosmedicine.org

1. Please revise your title according to PLOS Medicine's style. Your title must be nondeclarative and not a question. It should begin with main concept if possible. "Effect of" should be used only if causality can be inferred, i.e., for an RCT. Please place the study design ("A randomized controlled trial," "A retrospective study," "A modelling study," etc.) in the subtitle (ie, after a colon).

2. Abstract: Please structure your abstract using the PLOS Medicine headings (Background, Methods and Findings, Conclusions; Please combine the Methods and Findings sections into one section, “Methods and findings”)

3. Abstract: Please provide the dates of search, types of study designs included, eligibility criteria including language of publication. Please briefly mention the countries represented in the studies, as well as some information on types of included studies, participants, and relevant characteristics of studies.

4. Abstract: Please remove italics used for emphasis.

5. Abstract: As the final sentence of the Methods and Findings section, please describe the main limitation(s) of the study's methodology.

6. Abstract: Conclusions: In the second sentence, we suggest phrasing this as the overall finding of your study, for example: “In this study, we observed that few package combinations showed evidence of effectiveness.” or similar.

7. Author Summary: At this stage, we ask that you include a short, non-technical Author Summary of your research to make findings accessible to a wide audience that includes both scientists and non-scientists. The Author Summary should immediately follow the Abstract in your revised manuscript. This text is subject to editorial change and should be distinct from the scientific abstract. Please see our author guidelines for more information: https://journals.plos.org/plosmedicine/s/revising-your-manuscript#loc-author-summary

8. Throughout: Please use numbers within brackets for in-text reference citations, like this [1]. Please avoid the use of italics for emphasis.

9. Introduction: Page 5, Lines 9-10: Please temper this with “... to the best of our knowledge,to date there has been no synthesis to evaluate…” or similar.

10. Methods: Page 5: Line 24: Please include the sentence "This study is reported as per the Preferred Reporting Items for Systematic Reviews and Meta-Analyses (PRISMA) guideline (S1_Checklist)."

11. Methods: Page 6, Lines 5-6: Please update your search to the present time.

12. Methods: Page 6-7: It would be preferable to format the description of eligibility criteria for included studies in paragraph form, rather than as an outline/list.

13. Methods: page 7, Lines 8-9: Please ensure that all abbreviations are fully spelled out at their first appearance in the main text (for example, UCTs, CCTs)

14. Methods: Page 8: Please comment on assessment of publication bias.

15. Methods: Page 9, lines 9-10: Please clarify the description of correction for clustering- “The analysis corrected for clustering to address possible unit of analysis error”

16. Methods: Page 10, line 6-7: “Eight protocols were identified that could meet the criteria for

7 inclusion in an update to the review.” Was an attempt made to contact the authors to determine if the study findings have since been published? 

17. Results: Page 15, Line 7-8: Please present separate the results for the randomized vs. non-randomized studies, if possible.

18. Results: Throughout (for example page 23, line 19) please give the exact p value rather than p<0.05 for the meta analysis results (however please report p<0.001 when appropriate).

19. References: Please use the "Vancouver" style for reference formatting, and see our website for other reference guidelines https://journals.plos.org/plosmedicine/s/submission-guidelines#loc-references

20. PRISMA Checklist: Thank you for including the PRISMA checklist. Please update the checklist, using section and paragraph numbers to refer to locations within the text, instead of using page numbers.

21. Figures 2, 3, 4: For the forest plots, please present the overall result from the meta-analysis, with 95% CI and p values.

22. Table 1: Please note in the legend that “CT” stands for cash transfer.

Comments from the reviewers:

Reviewer #1: See attachment

Michael Dewey

Reviewer #2: This paper explores an interesting topic by systematically examining results from published studies about cash transfers targeting infants or young children development outcomes. The originality of this study, compared to previous published reviews of the impact of cash transfers, is to address the delicate question of cash transfers combined with an additional intervention targeting child development and to explore whether this combination (named "Cash+") gives better results than one of the components alone (cash alone or "plus" alone). This kind of information is really needed to help decision makers, planners and various stakeholders design the best interventions possible in each particular context.

The study is rigorously conducted, the analysis includes meta-analyses when doable and the paper is well written. However, due to the small number of studies available, the variety of contexts and a large heterogeneity of the studies included in the review, both in terms of outcomes measured, in the kind of "plus" interventions and in the intervention designs, results are a bit disappointing. In the end, very limited evidence can be shown and no firm conclusion can be drawn from this work. The authors are not to be blamed for this, of course, and it's obvious that they did their best to assemble the available information, but the practical impact of such a study remains limited. 

One wonders if other choices could have been made to draw more information from the corpus of 63 articles that were assessed for eligibility, by applying less strict inclusion and/or exclusion criteria. It is out of the scope of the review of this paper to fully examine all 43 excluded articles, and I know that when undertaking such a systematic review the right balance is really difficult to establish between being very inclusive (and having to deal with more heterogeneity) or very exclusive (and ending up with a too few number of studies). Nevertheless, it seems to me that some criteria could have been less strict in order to increase the number of studies to be included, at least in the narrative analysis (the qualitative synthesis). 

- For example, they considered as outcomes only what they call "third order SDG-indicators" related to children <5y old. The rationale behind this choice is not well explained and doesn't sound as the most effective choice. Indeed, when discussing their results, the authors themselves note that reaching "distant" outcomes is far more difficult than more proximal ones (discussion section, page 2 lines 10-18). Why not, then, having considered more proximal outcomes? Indeed, in this part of the discussion the authors could have elaborated a bit more about what implications does that have; in particular because many authors tend to think that there has been a too strong focus on ultimate (nutritional indices) outcomes in recent years, while many interventions achieve good results on more proximal outcomes. Also, to me the reference to SDGs is not very appropriate since SDGs target populations as a whole, not necessarily individuals (e.g. SDG targets 1.1.1 to 1.2.2, 2.1.1, 3.2.1, 3.2.2, 3.3.1-3.3.5… that the authors mentioned in their paper are all SDG indicators that are measured at the population level). 

- Also, the authors restrained the selection of studies to a list of "plus" components, referring again to the SDGs. First, in their protocol published on Prospero the authors mentioned SDG6 (water & sanitation) as one of the SDGs considered (Supplement 7, page 2) but they didn't mention that in the list of considered SDGs in the paper. If there was no study adding a WASH component to a cash transfer, which I doubt, they should at least have said so. Besides this, anyway, I think that referring to conceptual models of the young child development would have been a far better option than the SDGs (that don't have any conceptual model behind; and are not specifically targeting the child development per se). 

- Still in terms of studies selection, the authors could also have thought back of the final objective of their work, which might have been to give useful information for planners rather than highlighting a lack of evidence on a narrower question: indeed, by strictly focusing on interactions between cash and plus components they excluded studies with a 'cash-alone' and a 'cash+' arms, but without a 'plus alone' arm. At the end of the day, this is likely to have prevented them to examine more in-depth the impact of cash+ versus cash-alone interventions, which is a very interesting question. Interestingly enough, the first sentence in the conclusions says "Questions in social protection have remained as to what complementary interventions are necessary for cash transfers to have maximal impact" (page 31, lines 1-2). Which is exactly my point, but not exactly the question they addressed in their review.

- I also checked one or two excluded studies and failed to find an obvious reason for exclusion, even with the strict criteria applied in the study (e.g. the first in the alphabetical order: Adubra et al. 2019). 

Again, I know how harsh it is to conduct such a systematic analysis. All the above is not to say that the study was badly conducted (which is not the case) but I wanted to highlight that other choices could have led to a more informative review. This can't be for sure, of course, but this deserves at least to be discussed quite extensively in the discussion section.

Another major point I'd like to make regards the absence of information about the evaluation designs. While there is information about duration, age range, etc., it's obvious when reading the abstracts of the included studies that some of them used longitudinal designs (same children are followed-up and surveyed at baseline and endline) while others are based on repeated cross-sectional surveys at baseline and endline, on children of the same age range. This makes a huge difference in terms of interpretation, potential biases, duration of exposition to the intervention (for repeated cross-sectional designs) and statistical power. I found it very strange that this was not accounted for in the study and that this point is not even mentioned I the discussion section. 

Also, regarding studies for which the outcome is the child undernutrition, the authors analysed the impact in terms of rates of undernutrition. This is not the most powerful analysis, in particular for studies using a longitudinal evaluation design. Why not analysing the results in terms of changes of the continuous underlying variables (e.g. height-for-age or weight-for-length/height z-scores)? 

My last major comment regards the discussion section. The paper is already quite long; results are well and quite extensively described and the discussion starts with a summary of the results (page 25), which is good. As said above, and as acknowledged by the authors, at this point we don't have much evidence, no firm conclusions, nor much information we don't already know. I understand that the authors made some efforts to comment about some characteristics of cash or cash+ interventions that could increase impacts on some outcomes. But let's face it: they don't have much to say that really comes from their study. 

- About the context, for example, they draw some comments on the role of food transfers (page 26, first paragraph) but all what that says is that in emergency contexts, when dealing with acute undernutrition in infants and young children, it is key to give access to some nutritious foods (that are indeed most often ready-to-use supplementary foods) on top of the cash transfer. We already know that but it's good to have some combined evidence. However, the rest of the paragraph is made of more general statements that are not really related to the work presented. 

- Similarly, when commenting on the importance of the age of children for interventions targeting stunting (page 26, lines 14-18), the authors refer to the "wider literature in cash transfers" while in fact this has been demonstrated years ago in the nutrition literature. And in a few other parts of the discussion, the authors err on the side of making general comments that are not directly related to their results. This is likely because results are poor, of course, but then the discussion can be shorter. 

- The first paragraph of the recommendations for research, again, describes good but already well known points about cash or + intervention components, but those statements don't come specifically from this review. 

Minor comments:

- Please avoid the term "malnutrition" when it obviously refers to "undernutrition" (which is the preferred term, since "malnutrition" encompasses also overweight, obesity and related noncommunicable diseases). Page 7 line 18, on the contrary, "malnutrition" should replace "malnourishment".

- Please note also that "underweight" is not a type of undernutrition (such as stunting and wasting); it is, indeed, only an anthropometric indicator of one of the two types above, or both. 

- UCT and CCT acronyms are used without being defined

- Page 10, lines 8-9: this last sentence of the paragraph belongs to the methods section (where it is already said: page 9, line 5) but not to the results section. 

- Page 11, lines 3-4: "CCTs (7 studies), UCTs (11 studies), and public works programmes (3 studies)" add up to 21 studies, not 20.

- Page 11, lines 7-9: even if the information is in the table, it would be good to have the number of studies for each category of 'plus' components.

Reviewer #3: This is a timely and important topic, as more countries and development partners increasingly implement cash plus or integrated programming to address multidimensional poverty. This paper is well motivated and the background is well written. I have the following comments to improve the manuscript.

1. I find the comparison to either cash alone or plus alone somewhat problematic. If the authors stick to cash plus v. cash only, then you can make conclusions about whether 1) the plus component is effective or 2) "cash plus" is more effective that cash alone (if there are no differences you would conclude it is not). However, the combination of comparing cash plus to either cash alone or plus alone makes it difficult to understand what the conclusion is. I agree with their decision to do meta-analysis on cash plus v. cash only (avoiding mixing in of plus only with cash only as a comparison) but question the descriptive inclusion of cash plus to plus only. In an ideal world, you would want to compare studies that had the following three arms: 1) cash only, 2) plus only, 3) cash plus and 4) controls. This would allow you to understand impacts of "cash plus" and the synergies of cash and plus components (above and beyond either component separately). However, the authors could not do this because a sufficient number of studies with this design does not exist. Thus, the authors had to make a choice and in the background/framing, they seem to choose not to understand overall impacts of cash plus, but rather they start with cash as "standard of care" and seek to understand whether cash plus is more effective than cash alone. Following from this logic and the theoretical framing, they should drop the comparison to plus only, which just tells you whether cash is effective by itself or whether the combination is more effective than services/other interventions alone.

2. Page 21: Lines 6-11: It's not clear whether the discussion of heterogenous impacts follows the current paper's model of cash plus v. cash only, or whether they are citing findings from the original studies being reviewed, which may have compared to a pure control, and therefore should be removed from the results section.

3. This line seems out of place and doesn't connect to the sentence or larger paper: "one study emphasises that the burden of high fertility rates puts pressure on already strained social services in meeting the demands for early child development (Premand & Barry, 2020)."

4. The language on page 26, line 27 should be tempered. While the studies examined are rigorous, there are still a very limited number of studies examining cash plus v. cash alone and therefore the ability to generalize is still limited until more studies are published.

5. In the comparison to Manley et al. (2020), can the authors add a sentence to explicitly state how this study adds to the literature (i.e., what the combined findings suggest)? They appear to be saying that since Manley et al. found impacts of cash plus or cash plus on stunting, and the current authors find no difference between cash only and cash plus, that cash alone appears to be driving the impacts. It would help if the authors could clarify their conclusion.

6. This conclusion should be tempered and contexualized within the limitations of the study: "Findings also suggest limited or no added impact of nutrition behaviour change and psychosocial stimulation interventions and possibly worsening inequality from some cash-plus combinations compared to cash alone." In fact, Roy et al. (2019) have found that in terms of sustained impacts in Bangladesh, transfers + BCC had sustained impacts whereas transfers only did not on several pathways and outcomes important for child health (i.e., maternal IPV, economic resources, agency, social control and household poverty). The limitations of the current paper should be more clearly stated, that they are simply comparing effectiveness of the "plus" components and in fact have not compared cash v. control as compared to cash plus v. control. The statements/conclusions they are making would require the latter, which the authors have not done.

[LINK]

---

## [Decision Letter · Decision Letter 2]

11 Jun 2021

Dear Dr. Little,

Thank you very much for re-submitting your manuscript "Effectiveness of cash-plus programmes on early childhood outcomes compared to cash transfers alone: A systematic review and meta-analysis in low- and middle-income countries." (PMEDICINE-D-21-00432R2) for consideration in PLOS Medicine’s Special Issue: Global Child Health: From Birth to Adolescence and Beyond.

I have discussed the paper with my colleagues and the academic editor and it was also seen again by three reviewers. I am pleased to say that provided the remaining editorial and production issues are dealt with we are planning to accept the paper for publication in the journal.

[LINK]

We look forward to receiving the revised manuscript by Jun 18 2021 11:59PM.   

Sincerely,

Caitlin Moyer, Ph.D.

Associate Editor 

PLOS Medicine

plosmedicine.org

Requests from Editors:

1. Abstract: Please define the abbreviation “SDG” and “LMIC” at first use.

2. Abstract: Methods and Findings Line 26-27: Please provide the result for “...but had no added impact in improving weight-for-height or weight-for-age”

3. Author summary: Please rename the “non-technical summary” as “Author Summary”

4. Author summary: What do these findings mean?: In the first point, please qualify this with “There are few studies to date…” or similar. In the second point, please be more specific or clarify what is meant with “...improving the lives of vulnerable children.“

5. Introduction: Page 5 Line 9: We suggest revising this sentence to clarify/replace the word pernicious.

6. Results: If feasible, please include a sensitivity analysis omitting the 3 non-randomized studies.

7. Results: Page 17: Here and throughout the text, please use person first language where possible (avoid referring to children as “stunted” or “wasted”).

8. Results: Page 20 Line 1-2: Please revise to : “...were no longer statistically significant four years post-programme.” or similar.

9. Results: Page 24 Line 18: We would suggest providing the specific reference here, to clarify.

10. References: Please check to ensure that the "Vancouver" style is used for reference formatting (including Journal Title abbreviations, for example PLOS Med should be PLoS Med for reference 11), and see our website for other reference guidelines https://journals.plos.org/plosmedicine/s/submission-guidelines#loc-references

11. Supporting information files: Please rename/refer to files as S1 text, S1 Checklist, S1 Table, etc. or similar. On page 31 where the files are listed, please provide descriptive titles and legends, where appropriate.

12. PRISMA checklist: Please remove all references to page numbers (please refer only to section and paragraph locations).

13. Supplement 3: Please more clearly indicate the study designs and/or otherwise indicate the three non-randomized studies.

14. Supplement 7: Please provide a clean version of this document (without the comment).

Comments from Reviewers:

Reviewer #1: The authors have met most of my points. It is hard to believe that Google Translate could not cope with the article not in English nor, assuming it is the one in Spanish, that nobody at any of the institutions to which the authors are affiliated can read Spanish but if the authors wish to leave it out there is nothing much to be done.

Michael Dewey

Reviewer #2: This paper has been greatly improved as compared to the previous version. The comments I made have been satisfactorily taken into account. In particular, the discussion section is much more concise and "to the point". The authors have to be commended also for having to re-run analyses. Good job !

I have only a few additional (very)minor comments to make:

- In the abstract (page 2 line 5) then in the introduction section (page 5 lines 18-19) there are two expressions that are not readily understandable: "individual behavioural mediators" and "supply-side moderators". Indeed, the explanation of their meaning comes only in the result section (page 13, lines 2 and 4). I suggest to re-write (e.g. "improved mediating outcomes and/or enhanced supplies")

- In the last bullet of the non-technical summary, I suggest to explicitely mention the main limitation of the paper, i.e. the small number of studies that were available for analysis (despite all efforts); indeed, saying "more research is needed" is not enough, in my opinion (and can be said for almost all papers!)

- The expression "caloric-food transfer" (page 7 line 27 and page 21 line 6) is not a standard one and has no real meaning from a nutritional point of view; indeed, all foods are "caloric" by nature (except water); if the authors want to insist that the food-transfer used products that are high in calories, the correct expression would be "energy-dense foods", but even that wouldn't be correct regarding the 3 studies that are analyzed, since at least one of them used "normal" foods (not specificaly designed energy-dense products). Please replace by "food transfers".

- Page 20, line 20, the Calvo-Dercon Poverty Score is not known by everyone; adding a bibliographic reference would be welcome.

Yves MARTIN-PREVEL

Reviewer #3: The authors have addressed my concerns and I recommend publishing.

[LINK]

---

## [Editor Report · Decision Letter 3]

15 Jun 2021

Dear Dr Little, 

On behalf of my colleagues and the Academic Editor, Zulfiqar Bhutta, I am pleased to inform you that we have agreed to publish your manuscript "Effectiveness of cash-plus programmes on early childhood outcomes compared to cash transfers alone: A systematic review and meta-analysis in low- and middle-income countries." (PMEDICINE-D-21-00432R3) in PLOS Medicine’s Special Issue: Global Child Health: From Birth to Adolescence and Beyond.

PRESS

Sincerely, 

Caitlin Moyer, Ph.D. 

Associate Editor 

PLOS Medicine